# Deconstructing Immune Cell Infiltration in Human Colorectal Cancer: A Systematic Spatiotemporal Evaluation

**DOI:** 10.3390/genes13040589

**Published:** 2022-03-25

**Authors:** Emese Irma Ágoston, Balazs Acs, Zoltan Herold, Krisztina Fekete, Janina Kulka, Akos Nagy, Dorottya Mühl, Reka Mohacsi, Magdolna Dank, Tamas Garay, Laszlo Harsanyi, Balazs Győrffy, Attila Marcell Szasz

**Affiliations:** 1Department of Surgery, Transplantation and Gastroenterology, Semmelweis University, 1082 Budapest, Hungary; emeseagoston@gmail.com (E.I.Á.); fekete.krisztina@med.semmelweis-univ.hu (K.F.); harsanyi.laszlo@med.semmelweis-univ.hu (L.H.); 2Department of Oncology and Pathology, Karolinska Institutet, 171 77 Stockholm, Sweden; acs.balazs@med.semmelweis-univ.hu; 3Department of Clinical Pathology and Cancer Diagnostics, Karolinska University Hospital, 171 77 Stockholm, Sweden; 4Division of Oncology, Department of Internal Medicine and Oncology, Semmelweis University, 1083 Budapest, Hungary; herold.zoltan@med.semmelweis-univ.hu (Z.H.); muhl.dorottya@med.semmelweis-univ.hu (D.M.); mohacsi.reka@med.semmelweis-univ.hu (R.M.); magdolna.dank@med.semmelweis-univ.hu (M.D.); garay.tamas@itk.ppke.hu (T.G.); 52nd Department of Pathology, Semmelweis University, 1091 Budapest, Hungary; kulka.janina@med.semmelweis-univ.hu; 6Department of Pathology and Experimental Cancer Research, Semmelweis University, 1085 Budapest, Hungary; nagy.akos1@med.semmelweis-univ.hu; 7Faculty of Information Technology and Bionics, Pazmany Peter Catholic University, 1083 Budapest, Hungary; 8Department of Bioinformatics, Semmelweis University, 1094 Budapest, Hungary; gyorffy.balazs@med.semmelweis-univ.hu; 9Department of Tumor Biology, Korányi National Institute of Pulmonology, 1122 Budapest, Hungary

**Keywords:** colorectal cancer, immune checkpoint, lymphocyte, NanoString

## Abstract

Cancer-related immunity has been identified as playing a key role in the outcome of colorectal cancer (CRC); however, the exact mechanisms are only partially understood. In this study, we evaluated a total of 242 surgical specimen of CRC patients using tissue microarrays and immunohistochemistry to evaluate tumor infiltrating immune cells (CD3, CD4, CD8, CD20, CD23, CD45 and CD56) and immune checkpoint markers (CTLA-4, PD-L1, PD-1) in systematically selected tumor regions and their corresponding lymph nodes, as well as in liver metastases. Additionally, an immune panel gene expression assay was performed on 12 primary tumors and 12 consecutive liver metastases. A higher number of natural killer cells and more mature B cells along with PD-1^+^ expressing cells were observed in the main tumor area as compared to metastases. A higher number of metastatic lymph nodes were associated with significantly lower B cell counts. With more advanced lymph node metastatic status, higher leukocyte—particularly T cell numbers—were observed. Eleven differentially expressed immune-related genes were found between primary tumors and liver metastases. Also, alterations of the innate immune response and the tumor necrosis factor superfamily pathways had been identified.

## 1. Introduction

Cancer-related immunity is increasingly identified as an important complex factor in tumor initiation, growth, and progression [1]. Consequently, considerable research interest is focused on clarifying and on targeting the molecular mechanisms which are involved in cancer-related inflammation [2]. In the last decade, tumor infiltrating immune cells and cancer-specific cytokines have been proven as promising prognostic markers, however, their exact role in progression and their correlation between the different immune components involved have not been fully understood [3]. Based on the tumor infiltrating immune cells, a new scoring system was established, known as the ‘Immunoscore’. This has been providing promising prognostic information besides the classical TNM classification [4]. Immune scores obtained from intratumoral immune infiltrates have a strong prognostic impact in CRC. Furthermore, T-cell infiltration has been proven as an independent prognostic factor [5,6]. Increasing evidence suggests that CD3^+^, CD8^+^, and CD45RO^+^ cells play key roles in anti-tumor immune responses; furthermore, the number, type, and location of the infiltrating immune cells in primary tumors are a prognostic for disease-free survival and overall survival [5]. Immune-based new biomarkers could provide a wider therapeutic approach in cancer treatment. Microsatellite unstable (MSI) tumors have been observed to present extensive lymphocyte infiltration, a prominent T helper type 1/cytotoxic CD8^+^ T cell (Th1/CTL) immune microenvironment, and a high expression of checkpoint molecules such as programmed cell death protein 1 (PD-1, also known as CD279), programmed death-ligand 1 (PD-L1, also known as CD274), and cytotoxic T-lymphocyte-associated protein 4 (CTLA-4, also known as CD152) [7]. The blockade of these molecules to activate antitumor immunity has been proposed as a therapeutic strategy in CRC, and two clinical trials have been initiated to test PD-1 blockade in MSI CRC patients [8]. In this study, we performed a detailed analysis on selected CRC samples—tumor regions, lymph nodes, and metastatic tissues—to identify relevant correlations between the different immune cells and checkpoint molecules and their impact on prognosis.

## 2. Materials and Methods

A total of 242 primary CRC tumors and liver metastasis samples from 137 CRC patients diagnosed between 1987 and 2011 were evaluated in this study. Patients were randomly selected from the database of the 2nd Department of Pathology, Semmelweis University, Budapest, Hungary. The study was approved by the Regional and Institutional Committee of Science and Research Ethics, Semmelweis University (SE-TUKEB 207/2011). The clinicopathological characteristics of the patients are displayed in Table 1.

### 2.1. Immunohistochemical Staining of CRC Samples

Tissue microarrays (TMA) were composed from the formalin-fixed and paraffin-embedded (FFPE) samples with a systematic core punching algorithm using the Tissue Microarray Builder instrument (Histopathology Ltd., Pécs, Hungary). A total of 242 cores of 2 mm diameter were taken from the main tumor mass (MAIN) and/or liver metastasis (MET) of all patients. Additional samples from the tumor-normal interface (BORDER), deepest infiltrative area (FRONT), lymph node metastasis (LN), and normal colorectal mucosa (NORMAL) were also taken from 54 of the 137 patients.

Immunohistochemical reactions were performed on 4µm thick sections cut from TMA blocks mounted on adhesive glass slides (SuperFrost UltraPlus from Gerhard Menzel Ltd., Braunschweig, Germany). We analyzed the following biomarkers in each section: CD3 (pan-T cell marker), CD4 (helper T cell marker), CD8 (cytotoxic T cell), CD20 (B cell), CD23 (mature B cells, activated macrophages), CD45 (leukocytes, LCA), and CD56 (natural killer cells) along with CTLA-4, PD-L1, and PD-1 immune checkpoint markers in an automated immunostainer (Ventana Benchmark XT, Roche, Tucson, AZ, USA) using the solutions and setting as provided by the manufacturer. The slides were digitalized with a Pannoramic P250beta slide scanner (3DHistech Ltd., Budapest, Hungary) and for the evaluation we used the Pannoramic Viewer with the support of the TMA and the Histoquant modules (3DHistech). Immunoreactions were measured in the lymphocytes regarding all reactions and in tumor cells in the case of CTLA-4, PD-L1, and PD-1. The reactions were quantified and calculated with computer-assisted image analysis by using the QuantCenter digital analyzer resulting in the number of positive cells per annotation where the size of each annotation corresponded to the core cylinder’s surface of 3.14 mm^2^.

### 2.2. NanoString Pan-Cancer Immune Panel

The RNA from 12 primary tumors and 12 liver metastases was obtained from FFPE tissue. The 24 samples were obtained as follows: 11 paired and additionally a primary tumor and a metastasis sample. Five 5-µm thick sections were cut from the FFPE blocks; and, by following the manufacturer’s instructions, total RNA was obtained with the High Pure FFPE RNA Isolation Kit (Roche, Basel, Switzerland). RNA concentrations were measured using the Qubit 4 Fluorometer (Thermo Fisher Scientific, Waltham, MA, USA). The RNA samples with adequate concentrations were hybridized to the nCounter^®^ PanCancer Immune Profiling Panel (NanoString, Seattle, WA, USA) containing 770 genes for 16 h using a thermocycler. The samples were transferred to the nCounter Prep Station (NanoString, USA) for further processing. The gene expression profiles of the samples were digitalized with the nCounter Digital Analyzer. The results were quantified using nSolver 4.0 Analysis Software (NanoString, USA).

### 2.3. Description of Clinicopathological Data

Tumor staging was given based on the grouping of the American Joint Committee on Cancer (AJCC) using histopathological examination of surgical specimens and imaging studies [9]. Sidedness of the tumor was described as follows. The tumor was grouped as right-sided if it was originating from cecum, ascending colon, and the proximal two-third of the transverse colon; and, it was considered as left-sided if it was originating from the distal one-third of the transverse colon, descending colon, sigmoid colon, and rectum [10].

### 2.4. Statistical Analysis

Data was analyzed within the R for Windows environment (version 4.1.2, R Foundation for Statistical Computing, Vienna, Austria). Due to multiple sampling from the same patients, group comparisons were performed using mixed effect linear models, where patient IDs were used as the random component (R-package nlme, version 3.1-155). In the case where a parameter had more than two factor levels, post-hoc comparisons were performed using the Tukey method. A *p* < 0.05 was considered as statistically significant.

NanoString data was analyzed using the RUVSeq method [11] (R-package RUVSeq, version 1.26.0 [12]). In short, after the normalization of the count data with RUVSeq, differential expression analysis (R-package DESeq2, version 1.32.0 [13]) and gene set enrichment analysis were performed. The prognostic role of NanoString count data on patient survival was analyzed using univariate and multivariate standard and mixed effect Cox regression models (R packages survival version 3.3-0 and coxme version 2.2-16). The results were drawn using volcano-plots and heatmaps with the ggplot2 [14] and ComplexHeatmap [15] R-packages, respectively.

## 3. Results

A total of 242 FFPE samples of 137 CRC patients were analyzed. A total of 139 (57.4%) and 103 (42.6%) of the 242 samples originated from the MAIN and MET sites, respectively. A comparison of the number of immune-based biomarker-positive and immune checkpoint marker-positive cells was performed using the following grouping factors: type (MAIN vs. MET), sex, sidedness, lymph node status, and AJCC staging; in the latter, stages I and II were pooled.

A significantly higher number of CD56^+^ cells were observed in the MAIN samples (*p* = 0.0195, Figure 1A and Appendix A) compared to those of the MET samples. While CD23^+^ (*p* = 0.1133, Figure 1B) and PD-1^+^ (*p* = 0.1312, Figure 1C) cells occurred more tendentiously in the MAIN samples, no difference could be justified in the case of CD3^+^, CD4^+^, CD20^+^, CD45^+^, CTLA-4^+^, and PD-L1^+^ cells between the MAIN and the MET samples.

Further group comparisons were examined separately on the MAIN and the MET datasets. Within the MAIN samples, the number of PD-1^+^ cells were significantly higher (*p* = 0.0092, Figure 2A), and a tendentiously higher number of CD45^+^ (*p* = 0.1313, Figure 2B) cells were found in right-sided tumors. A higher number of metastatic lymph nodes were associated with significantly lower CD20^+^ (N0 vs. N2: *p* = 0.0119, N1 vs. N2: *p* = 0.0292, Figure 3A). Moreover, more advanced lymph node metastasis status was associated with marginally and tendentiously higher CD3^+^ (N1 vs. N2: *p* = 0.0587, Figure 3B) and CD45^+^ (*p* = 0.1204, Figure 3C) cell counts, respectively. No differences between the MAIN samples were found related to the sex and the AJCC staging of patients. Mean ± standard error results from all of the comparisons are summarized in Appendix A.

In the MET samples, we found differences only within CD56^+^ and CD45^+^ cell counts: CD56^+^ was lower in those cases, where the metastasis was present since the diagnosis of CRC (Stage I–II vs. IV: *p* = 0.0208, Stage I–II vs. III: *p* = 0.1056; Figure 4A); similarly, CD45^+^ was marginally higher in Stage IV CRC (Stage III vs. IV: *p* = 0.0820; Figure 4B). Mean ± standard error results from all of the comparisons are summarized in Appendix A.

### 3.1. Distribution of Markers in Different Areas of CRC and Metastatic Lymph Nodes

Of the 137 patients, 54 were available for further immune-based biomarker and immune checkpoint marker distribution analysis. Immune cell distribution was evaluated in the following regions: normal colon tissue (NORMAL), main tumor mass (MAIN), tumor-normal interface (BORDER), deepest infiltrative area (FRONT), and lymph node metastasis (LN). A clinicopathological description of this study subpopulation is summarized in Appendix A.

The markers CD4^+^ and PD-L1^+^ showed no significant difference in any of the investigated regions. However, CD56 showed a higher expression in normal colon tissue compared to the other regions (*p* < 0.0001). CD3, CD8, CD20, CD23, CD45, CTLA-4 and PD-1 showed significantly higher expression in lymph node metastases only (*p* < 0.0001 compared to all other sites, except CTLA-4: *p* = 0.0008 vs. BORDER, *p* = 0.0005 vs. FRONT, *p* = 0.0004 vs. MAIN, *p* = 0.0021 vs. NORMAL; and PD-1: *p* = 0.0022 vs. BORDER; Figure 5 and Appendix A).

### 3.2. Immune Panel Gene Expression Analysis

A total of 12 MAIN and 12 MET samples were selected for analysis with the NanoString nCounter^®^ PanCancer Immune Profiling Panel. The samples originated from 13 patients, whose clinicopathological description was summarized in Appendix A. Differentially expressed genes (DEGs) were identified using differential expression analysis. Between MAIN and MET samples, 11 and 29 DEGs were, respectively, identified as significantly and marginally different (Figure 6, Appendix A). Of the 11 DEGs, the complement C4B (*C4B*), complement factor I (*CFI*), defensin beta 1 (*DEFB1*), interleukin-1 receptor accessory protein (*IL1RAP*), interleukin-27 (*IL27*), mannose binding lectin 2 (*MBL2*), and metallophosphoesterase domain containing 1 (*MPPED1*) genes were downregulated. While the caspase recruitment domain family member 9 (*CARD9*), C-C motif chemokine receptor 7 (*CCR7*), lymphotoxin beta (*LTB*), and tumor necrosis factor (TNF) receptor superfamily member 8 (*TNFRSF8*) genes were upregulated (Appendix A). No difference was found in the gene expression profile of right-sided and left-sided tumors (Appendix A).

Gene over-representation was investigated via gene set enrichment analysis (GSEA). Decreased expression of innate immune response genes (6 of the 11 DEGs, odds ratio (OR): 16.04, 95% confidence interval (CI): 2.44–∞, *p* = 0.0133) was seen. The pathways related to the TNF superfamily members and their receptors were also altered, marginally increased expression of *LTB* and *TNFRSF8* was observed (OR: 24.38, 95% CI: 2.56–∞, *p* = 0.0659). No further alternated pathways were identified (Figure 7).

The prognostic significance of the identified DEGs was examined using two types of survival models. First, the 12 MAIN and the 12 MET samples were analyzed separately using standard Cox regression models. Second, all of the 24 samples were analyzed in a mixed effect Cox regression model, where patient’s IDs and sample source were used as the random and as the stratification factor, respectively. Standard univariate Cox regression results showed that the disease-specific survival (DSS) of patients was affected by lower MAIN *TNFRSF8* counts (*p* = 0.0378), while the progression-free survival (PFS) was significantly affected by the higher MAIN *DEFB1* counts (*p* = 0.0410). Worse PFS of patients was associated with lower *C4B* (*p* = 0.0474), *CFI* (*p* = 0.0449), and *IL1RAP* (*p* = 0.0266) counts and higher *CARD9* (*p* = 0.0224) counts within the MET samples. Stratified, mixed effect models revealed no further results (Appendix A).

No significant differences were found via the standard multivariate survival models, while the following was observed using the second method: *C4B* (*p* = 0.0240), *MBL2* (*p* = 0.0180), *CARD9* (*p* = 0.0160), and *TNFRSF8* (*p* = 0.0093) significantly affected DSS; *IL27* (*p* = 0.0220) and *LTB* (*p* = 0.0350) were prognostic for PFS, while *DEFB1* had a significant effect both on DSS (*p* = 0.0390) and PFS (*p* = 0.0420). Hazard ratios and 95% confidence intervals are shown in Appendix A.

## 4. Discussion

Despite a considerable number of studies, personalized therapy in CRC remains challenging. The heterogeneity of colorectal cancer has been readily identified and understood in clinical decision making at epigenetic, genetic, transcriptomic, proteomic, and microenvironmental levels [16]. Genomic alterations during carcinogenesis are the drivers of cancer progression in the metastatic setting as well. This is an applied phenomenon in the clinical setting when we utilize RAS mutation status to define the patient group refractory to epidermal growth factor receptor (EGFR) monoclonal antibodies. Additionally, BRAFV600E mutations associate with poor prognosis, and they predict decreased sensitivity to standard therapies, implying targeted combinations incorporating BRAF inhibitors in practice. Sequential analysis of circulating tumor DNA has the potential to guide future therapeutic decisions in CRC [17]. Undoubtedly, transcriptomic subtypes and pathway activation signatures have also shown prognostic and potential predictive value in metastatic CRC, reflecting stromal and immune microenvironment interactions with cancer cells [18]. The MSI CRC patients’ tumors are known to be sensitive to immune checkpoint inhibitors, whereas for those with the mesenchymal phenotype resistant immunosuppressive cascades are applied.

In our study, we aimed at assessing the immune cell infiltration of CRC in systematically selected tumor regions spatiotemporally in 137 patients. We evaluated the spectrum of the tumor progression route from the location of the primary tumor toward the invasive front through lymph node metastases, ending up eventually in the liver. We analyzed leukocytes in general, T cells, helper T cells, cytotoxic T cells, B cells, mature B cells/activated macrophages, and natural killer cells, along with CTLA-4, PD-L1 and PD-1 immune checkpoint markers in each location. According to relatively recent data, the pattern of immune cell infiltrates is prognostic and predictive for outcome and response to therapies [19]. It also seems now evident to assess them in the primary tumors and liver metastases of CRCs separately [20]. We observed a higher number of natural killer cells in the main tumor area as compared to metastases. This is in line with the observation that natural killer cells are one of the gatekeepers of gut carcinogenesis, and they eventually get exhausted with tumor progression [21]. More mature B cells along with PD-1^+^ expressing cells occurred in the main tumor mass, while no difference was observed in T cells and their subpopulations or the CTLA-4^+^ and the PD-L1^+^ cells between primary and metastatic samples. Based on our findings, B cells in the primary tumor display a reaction to the initially formed tumor cell population and its local consequences, and eventually in the metastasis they are less apparent players in the interaction. This is in line with the observation that B cells are members of the surveillance system overlooking the gut microbiome and neoplastic steps, as they likely play an indirect role by recruiting T cells to the tumor while forming islets and producing immunoglobulins [22]. Additionally, B cells in a higher density reflected better prognosis in right-sided CRC [23].

A higher number of metastatic lymph nodes were associated with significantly lower B cell counts. The more advanced the lymph node metastatic status, the higher the leukocyte and particularly T cell numbers were observed. In a recent publication, T reg cells were found in a higher amount with increased lymph node involvement, implying a role for this subpopulation in facilitation of tumor progression [24,25].

Within the liver metastases, we found differences in natural killer and leukocyte counts: less natural killer cells were identified in patients presenting upfront with advanced stage disease (Stage I–II vs. IV) and leukocyte count was marginally higher in Stage IV CRC (Stage III vs. IV). M2-like tumor-associated macrophage infiltration is associated with an increased incidence of CRC liver metastasis and with promotion of disease progression in the liver. Additionally, TGF-β-induced epithelial-mesenchymal transition in cancer stem cells serves as a route for liver metastasis formation in CRC [26].

Helper T cell and PD-L1^+^ cell counts showed no significant difference in selectively investigated primary tumor regions. Natural killer cells were more prominently found peritumorally by us and others as well [27]. Leukocytes, including T cells, especially killer cells, B cells, and mature B cells as well as CTLA-4 and PD-1 expressing cells showed higher expression in lymph node metastatic tumor regions. This is not surprising, as lymph nodes serve as both communication and physiological spaces for immune cells and immune functions. In oncology, there were several approaches to understand this delicate interaction of the host system and the tumor. Even the number of surgically removed and pathologically analyzed lymph nodes is a quality measure, and it improves the outcome of CRC [28]. Inflammatory infiltration—including lymphocytes—in the primary tumor is not only prognostic, but also yields the number of lymph nodes harvested [29,30]. These observations are now confirmed by multiomic annotated datasets as well, providing additional evidence: high lymph node yield in colon cancer resections seem to be determined by a host–tumor adaptive immune response, and efforts to maximize the number of examined lymph nodes are somewhat cozening.

Available liver metastases with paired primary CRCs were investigated for gene expression of selected immune related genes with the NanoString method. Significantly 11 and marginally 29 DEGs were detected. Of the 11 DEGs, the complement C4B (*C4B*), complement factor I (*CFI*), defensin beta 1 (*DEFB1*), interleukin-1 receptor accessory protein (*IL1RAP*), interleukin-27 (*IL27*), mannose binding lectin 2 (*MBL2*), and metallophosphoesterase domain containing 1 (*MPPED1*) genes were downregulated. While the caspase recruitment domain family member 9 (*CARD9*), C-C motif chemokine receptor 7 (*CCR7*), lymphotoxin beta (*LTB*), and tumor necrosis factor (TNF) receptor superfamily member 8 (*TNFRSF8*) genes were upregulated. Others have found that the Wnt-beta-catenin pathway, the TGF-beta pathway, and various downstream activators of PI3K/AKT signaling are the most relevant mechanisms in CRC liver metastasis formation [31]. We, on the other hand, analyzed those 770 genes, which have been previously identified as relevant immune related factors. Apparently, no difference was detected in the immunologic gene expression profile of right-sided and left-sided tumors, however, we had a low number of tumors in each group. Gene set enrichment analysis revealed a decreased expression of innate immune response genes and the pathways related to the TNF superfamily members and their receptors were also altered (marginally increased expression of *LTB* and *TNFRSF8* were observed). A limited number of studies investigated signaling associated with liver metastasis formation in CRC; it was however documented that genes during tumor progression were strongly associated with either the cell adhesion/focal adhesion/chemokine signaling pathway/PI3K-AKT signaling pathway or innate immune response/complement activation/acute-phase response [32,33,34].

When analyzing the prediction of prognosis based on the genes’ expression in primary CRC, we found that a shorter DSS was associated with a lower *TNFRSF8* (*CD30*) expression, and a shorter PFS was seen with higher *DEFB1* counts. In liver metastases, shorter PFS was associated with a lower *C4B*, *CFI*, and *IL1RAP* expression and higher *CARD9* counts. In general, in colorectal carcinoma progression, *C4B*, *MBL2*, *CARD9*, and *TNFRSF8* affected DSS. While *IL27* and *LTB* were prognostic for PFS, *DEFB1* had a significant effect both on DSS and PFS. All of the above markers have been described in various cancer types, displaying a variable pattern of expression (*DEFB1*) [35], but with some therapeutic implications [36] in oncology or other specialties (*C4B*). In pro- and anti-tumorigenic inflammation in CRC, *CARD9* and leukotriene signaling are especially important, and they warrant further research [37,38] to better understand resistance mechanisms against immunotherapy in CRC and other solid tumors.

### Limitations of the Study

Limitations of our study include the following. A complete analysis of the distribution of immune-based biomarkers and immune checkpoint markers at all tumor regions was not performed for the whole study population. Similarly, gene analysis via NanoString could be performed only for a select number of patients. A further limitation of the study was that patient survival could be analyzed only for the NanoString subpopulation. The rationale behind the latter was that due to the large inclusion period, completely different treatment options were available for older and later patient enrollments, and those differences could cause significant bias in the proper evaluation of survival data. The patients of the NanoString subpopulation were homogenous in terms of time of diagnosis, surgery, etc.

## 5. Conclusions

Summarizing the results of our study, we observed a higher number of natural killer cells and more mature B cells along with PD-1^+^ expressing cells in the main tumor area. Metastatic lymph nodes were characterized by significantly lower B cell counts and higher leukocyte counts. Furthermore, 11 differentially expressed immune-related genes were found during the comparison of primary tumor and liver metastasis samples and significant alterations of the innate immune response and the TNF superfamily pathways were identified. The herein identified alterations in gene expression were associated with shorter survival times of CRC patients. Our results provide further insight into the immune cell composition of CRC, but further studies are needed to get more precise information on their exact contribution to tumor initiation and progression, and their possible role in therapeutic intervention.

## Figures and Tables

**Figure 1 genes-13-00589-f001:**
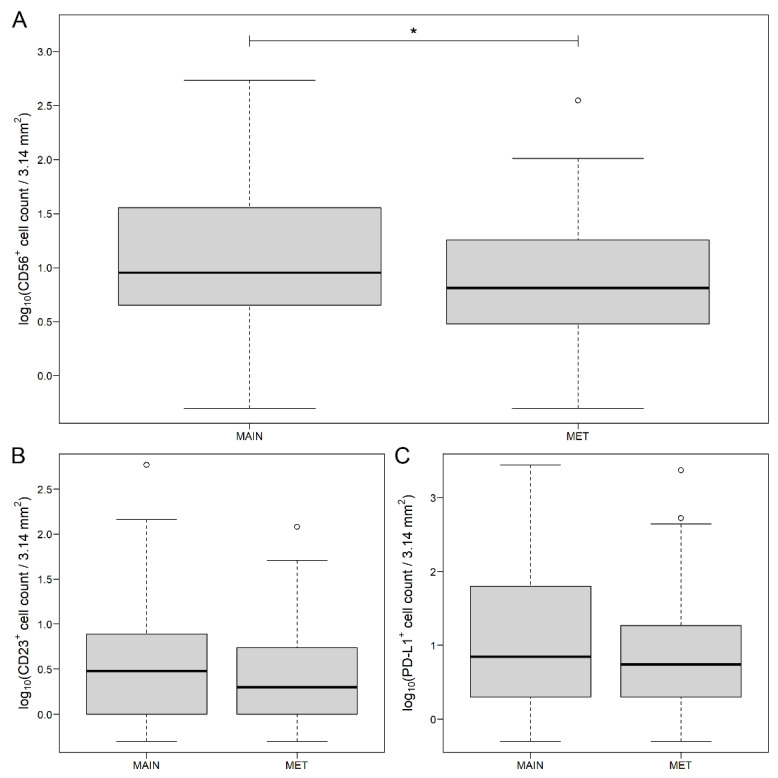
CD56 (**A**), CD23 (**B**) and programmed death-ligand 1 (PD-L1, (**C**)) count data of main tumor mass (MAIN) and liver metastasis (MET) samples of colorectal cancer patients. While CD56 was significantly different between the two sample types, only marginal differences were justified in the case of CD23 and PD-L1. The hollow black circles and the thick line represent outliers (>1.5 times the interquartile range above the upper quartile) and the median value, respectively. * *p* < 0.05.

**Figure 2 genes-13-00589-f002:**
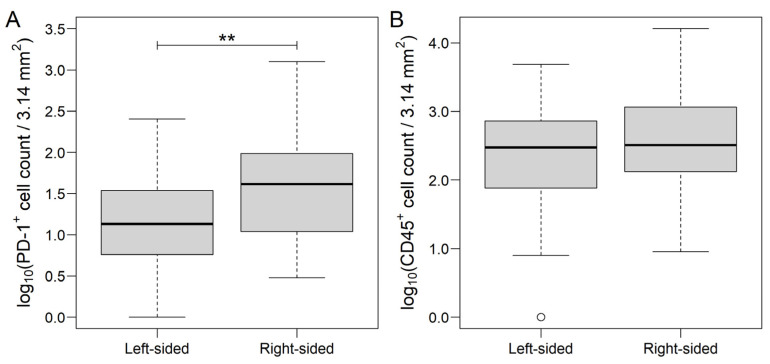
Programmed cell death protein 1 (PD-1, (**A**)) and CD45 (**B**) count data of left-sided and right-sided colorectal cancer samples. While PD1 was significantly different between the two sides, only marginal difference was justified in the case of CD45. The thick line represents the median value. ** *p* < 0.01.

**Figure 3 genes-13-00589-f003:**
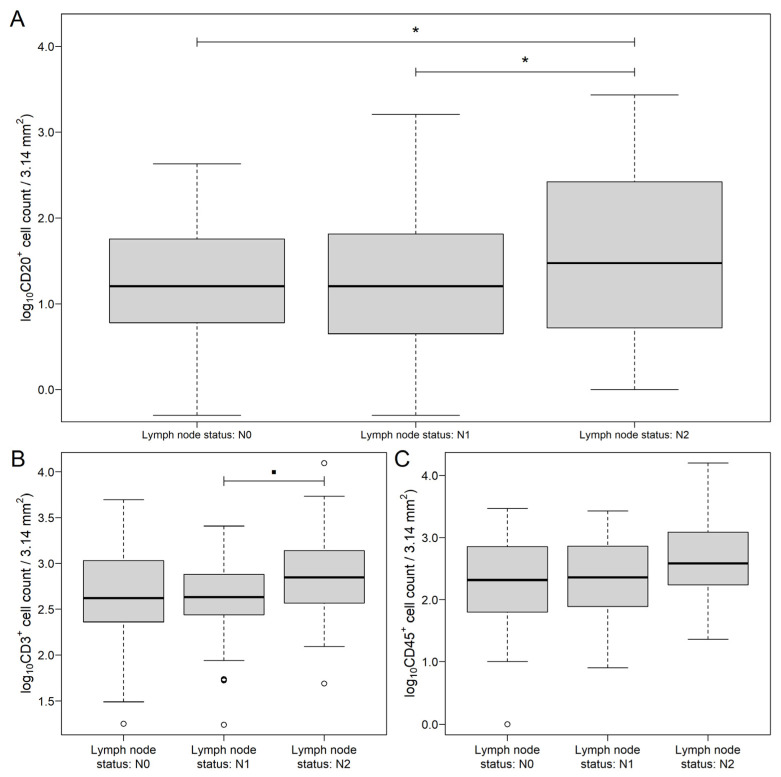
CD20 (**A**), CD3 (**B**) and CD45 (**C**) count data of colorectal cancer samples grouped by lymph node metastasis status. While CD20 was significantly different between the two sides, only marginal differences were justified in the case of CD3 and CD45. The hollow black circles and the thick line represent outliers (greater/less than 1.5 times the interquartile range above/under the upper/lower quartile) and the median value, respectively. * *p* < 0.05 and ▪ 0.1 < *p* ≤ 0.05.

**Figure 4 genes-13-00589-f004:**
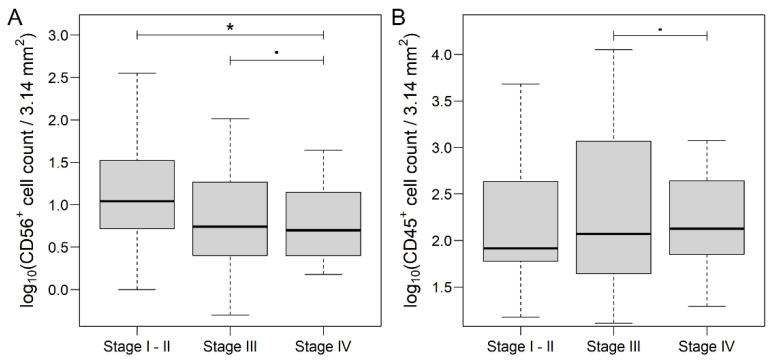
CD56 (**A**) and CD45 (**B**) count data of liver metastasis samples of colorectal cancer patients, grouped by AJCC staging [9]. CD56 count was significantly lower in those samples, where the metastasis was present at the time of tumor diagnosis simultaneously, while the opposite tendency was found in the case of CD45 counts. The thick line represents the median value. * *p* < 0.05 and ▪ 0.1 < *p* ≤ 0.05.

**Figure 5 genes-13-00589-f005:**
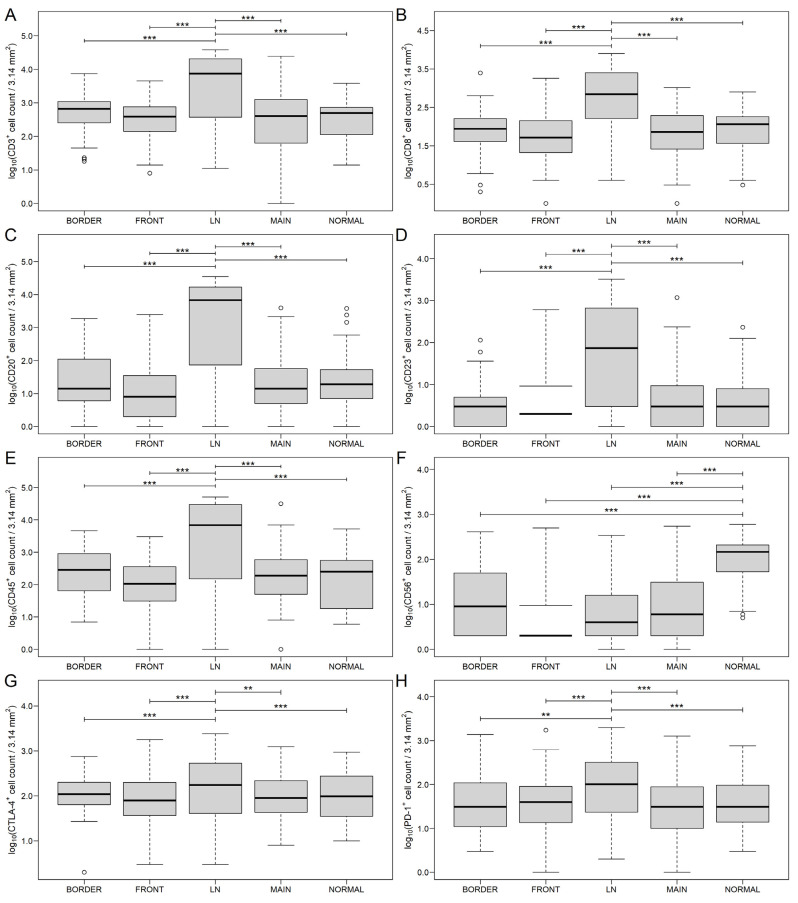
CD3 (**A**), CD8 (**B**), CD20 (**C**), CD23 (**D**), CD45 (**E**), CD56 (**F**), cytotoxic T-lymphocyte-associated protein 4 (CTLA-4, (**G**)) and programmed cell death protein 1 (PD-1, (**H**)) count data of colorectal cancer samples at different sites of the tumor. MAIN: main tumor mass; BORDER: tumor-normal interface; FRONT: deepest infiltrative area; LN: lymph node metastasis; and NORMAL: normal colon tissue. The hollow black circles and the thick line represent outliers (>1.5 times the interquartile range above the upper quartile) and the median value, respectively. ** *p* < 0.01, *** *p* < 0.001.

**Figure 6 genes-13-00589-f006:**
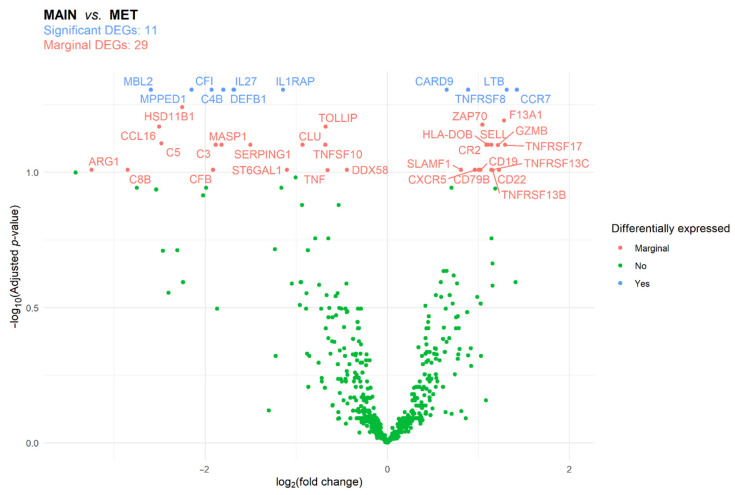
Differentially expressed genes (DEGs) between the main tumor mass (MAIN) and liver metastasis (MET) samples of colorectal cancer patients. The false discovery rate method was used for *p*-value adjustment.

**Figure 7 genes-13-00589-f007:**
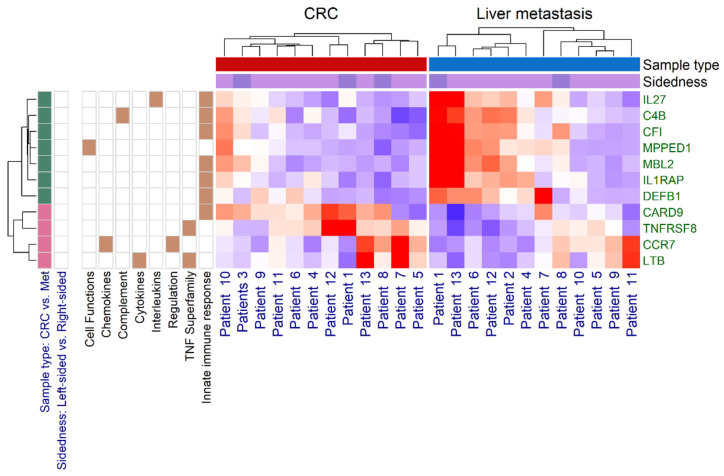
Heatmap of significantly different gene expressions between the main tumor mass (CRC) and liver metastasis (Met) samples of colorectal cancer patients. The green and pink boxes represent the downregulation and the upregulation of genes, respectively. Brown box shows enrichment annotation information of the differentially expressed genes.

**Table 1 genes-13-00589-t001:** Clinicopathological description of study population.

Parameter	Number of Observation/Mean ± SD
Age (year)	60.58 ± 11.01
Sex (Male:Female)	77:60 (56.2%:43.8%)
Location of the CRC ^1^	
– Coecum	16 (11.7%)
– Ascending colon	16 (11.7%)
– Transverse colon	11 (8%)
– Descending colon	15 (10.9%)
– Sigmoid colon	50 (36.5%)
– Rectum	28 (20.4%)
Sidedness of the tumor	
– Right-sided	45 (32.8%)
– Left-sided	92 (67.2%)
pT—extent of the tumor ^1^	
– T1	1 (0.7%)
– T2	11 (8%)
– T3	93 (67.9%)
– T4	27 (19.7%)
pN—lymph node status ^1^	
– N0	44 (32.1%)
– N1	48 (35%)
– N2	42 (30.7%)
AJCC [9] staging ^1^	
– Stage I	2 (1.5%)
– Stage II	27 (19.7%)
– Stage III	54 (39.4%)
– Stage IV	51 (37.2%)

^1^ No information about exact tumor location except for right-sided and staging was available for 1 and 3 patients, respectively. AJCC: American Joint Committee on Cancer; CRC: colorectal cancer.

## Data Availability

The datasets used and/or analyzed during the current study are available from the corresponding author on reasonable request.

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
