# Peer review of "Deconstructing Immune Cell Infiltration in Human Colorectal Cancer: A Systematic Spatiotemporal Evaluation"

_genes, 2022, doi:10.3390/genes13040589_

Round 1

Reviewer 1 Report

This is a concisely written manuscript built on sound experimental design and careful analysis of data. Strengths of the study include decent sample size (material from 137 patients was immunostained), automated and therefore unbiased image analysis, and a large panel of immune-related genes analyzed for differential gene expression. One shortcoming I would point out is the lack of clear conclusions about the prognostic/predictive significance of the findings. Suggestions on the practical applicability would be welcome in the Discussion which reads more like a reiteration of the Results than a synthesis. If the Authors think any of their findings may, e.g., facilitate the prediction of metastatic status from the features of the primary tumor, or inform the choice of therapy, this should be clearly stated. Such conclusions would greatly improve the potential clinical significance of the manuscript.

Minor comment / question: It is unclear whether the area of interest in the lymph nodes was limited to the area occupied by the metastasis. If not, demonstrating that immune cells (mostly lymphocytes) are more prevalent in a lymph node than in the primary tumors or the liver metastases is slightly too obvious. The conclusion in lines 281-282 (“Leukocytes, including T cells […] showed greater expression in lymph nodes”) is difficult to interpret if the analyzed area was not restricted to the metastasis.

English language requires minor revision here and there (e.g., "could have been" should be replaced by "were" at several instances), but otherwise the paper is fully comprehensible.

Reviewer 2 Report

Emese Irma Ágoston and colleagues presented the article entitled "A Deconstructing Immune Cell Infiltration of Human Colorectal Cancer: A Systematic Spatiotemporal Evaluation" that raises the issue of the molecular background of colorectal cancer (CRC) progression. They performed a detailed analysis of CRC samples: tumor regions, lymph nodes and metastatic tissues to identify the profile of immune cells and checkpoint molecules and their impact on CRC prognosis. Moreover, using the NanoString technology, they compared 770 immune-related genes expression between primary CRC and related liver metastases.

I found this article interesting and worth publishing, and it impacts the knowledge and brings us closer to personalized therapy of CRC.

The quality of the article is good, the experiment design was appropriate, and it is well written. Results are presented clearly, and they seem fine. I am not an English speaker, but the English quality looks quite good.

Nonetheless, there is an impression that the discussion section is too brief and the results insufficiently discussed against the existing knowledge - especially the section on results from comparative expression analysis between primary tumour and liver metastasis. There also lacks the summary conclusions.

I suggest that discussion should be little improved, particularly the newest reports on this topic.

I also have a few remarks:

  1. In lines 245-247, "The microsatellite instable CRC patients' tumors are known to be sensitive to immune checkpoint inhibitors, whereas those with mesenchymal phenotype are applying resistant immunosuppressive cascades." – there is a lack of a reference.
  2. Similarly, in lines: 269-271: "In a recent publication, T reg cells were found in higher amount with increased lymph node involvement, implying a role for this subpopulation in facilitation of tumor progression." – lacks a reference.
  3. In the sentence in line: 267, the authors stated: "Higher number of metastatic lymph nodes was associated with significantly lower B cell counts." And in the following: "The more advanced the lymph node metastatic status, the higher the leukocyte and particularly T cell numbers were observed". Then in line 282, they stated: "also Leukocytes, including T cells, especially killer cells, B cells and mature B cells and CTLA-4 and PD-1 expressing cells showed higher expression in lymph nodes."

That may be confusing. Weren't B cells lowered? May authors be more specific and precise here?

To summarise, Ágoston and colleagues did good work. I recommend this article to publish after slight improvement.

Reviewer 3 Report

The study is well design on recent thema of immune screening of CRC tumors. The topic is not so original and the results well presented. The discussion did not apport so orignal information.
